

# TCF7L2 rs7903146 polymorphism association with diabetes and obesity in an elderly cohort from Brazil

Lais Bride[1], Michel Naslavsky[2], Guilherme Lopes Yamamoto[2], Marilia Scliar[2], Lucia HS Pimassoni[3], Paola Sossai Aguiar[1], Flavia de Paula[1,4], Jaqueline Wang[2], Yeda Duarte[5,6], Maria Rita Passos-Bueno[2], Mayana Zatz[2] and Flávia Imbroisi Valle Errera[1,4]

[1] Biotechnology Graduate Program, Federal University of Espírito Santo, Vitória, Espírito Santo, Brazil
[2] Biosciences Institute, University of São Paulo, São Paulo, São Paulo, Brazil
[3] School of Science of Santa Casa de Misericórdia de Vitória, Vitória, Espírito Santo, Brazil
[4] Department of Biological Sciences, Federal University of Espírito Santo, Vitória, Espírito Santo, Brazil
[5] School of Nursing, University of São Paulo, São Paulo, São Paulo, Brazil
[6] School of Public Health, University of São Paulo, São Paulo, São Paulo, Brazil

Corresponding author
Flávia Imbroisi Valle Errera,
flavia.valle@ufes.br

## ABSTRACT

**Background:** Type 2 diabetes mellitus (T2DM) and obesity are complex pandemic diseases in the 21st century. Worldwide, the T allele rs7903146 in the *TCF7L2* gene is recognized as a strong GWAS signal associated with T2DM. However, the association between the C allele and obesity is still poorly explored and needs to be replicated in other populations. Thus, the primary objectives of this study were to evaluate the TCF7L2 rs7903146 association with T2DM according to BMI status and to determine if this variant is related to obesity and BMI variation in a cohort of elderly Brazilians.

**Methods:** A total of 1,023 participants from an elderly census-based cohort called SABE (Saúde, Bem Estar e Envelhecimento—Health, Well-Being and Aging) were stratified by BMI status and type 2 diabetes presence. The *TCF7L2* genotypes were filtered from the Online Archive of Brazilian Mutations (ABraOM—Online Archive of Brazilian Mutations) database, a web-based public database with sequencing data of samples of the SABE's participants. Logistic regression models and interaction analyses were performed. The BMI variation (ΔBMI) was calculated from anthropometric data collected in up to two time-points with a ten-year-assessment interval.

**Results:** The association between the rs7903146 T allele and T2DM was inversely proportional to the BMI status, with an increased risk in the normal weight group (OR 3.36; 95% CI [1.46–7.74]; $P = 0.004$). We confirmed the T allele association with risk for T2DM after adjusting for possible confounding variables (OR 2.35; 95% CI [1.28–4.32]; $P = 0.006$). Interaction analysis showed that the increased risk for T2DM conferred by the T allele is modified by BMI ($P_{interaction} = 0.008$), age ($P_{interaction} = 0.005$) and gender ($P_{interaction} = 0.026$). A T allele protective effect against obesity was observed (OR 0.71; 95% CI [0.54–0.94]; $P = 0.016$). The C allele increased obesity risk (OR 1.40; 95% CI [1.06–1.84]; $P = 0.017$) and the CC genotype

showed a borderline association with abdominal obesity risk (OR 1.28; 95% CI [1.06–1.67]; *P* = 0.045). The CC genotype increased the obesity risk factor after adjusting for possible confounding variables (OR 1.41; 95% CI [1.06–1.86]; *P* = 0.017). An increase of the TT genotype in the second tertile of ΔBMI values was observed in participants without type 2 diabetes (OR 5.13; 95% CI [1.40–18.93]; *P* = 0.009) in the recessive genetic model.

**Conclusion:** We confirmed that the rs7903146 is both associated with T2DM and obesity. The *TCF7L2* rs7903146 T allele increased T2DM risk in the normal weight group and interacted with sex, age and BMI, while the C allele increased obesity risk. The TT genotype was associated with a lesser extent of BMI variation over the SABE study's 10-year period.

# INTRODUCTION

Type 2 diabetes mellitus (T2DM) and obesity are considered pandemic diseases. They are interconnected by insulin mechanisms and characterized by complex interactions between environmental and genetic factors (*Haupt et al., 2010*; *Chen et al., 2018*; *Grant, 2019*). In this context, crucial T2DM genes involved in insulin production, processing, trafficking and secretion can also play a significant role in obesity development (*Noordam et al., 2017*; *Fernández-Rhodes et al., 2018*). The *TCF7L2* (10q25.2), one of these genes, encodes a transcription factor member of the Wnt signaling pathway known to act on vital functions of β cells and glucose metabolizing tissues (*Cropano et al., 2017*).

The rs7903146 T allele in *TCF7L2* is the strongest GWAS signal for T2DM risk in different populations across the world and it is associated with insulin synthesis, processing, secretion and action mechanisms (*Grant et al., 2006*; *Cauchi et al., 2008b*; *Bouhaha et al., 2010*; *Zhou et al., 2014*; *Corella et al., 2016*; *Cropano et al., 2017*). The genetic susceptibility for T2DM is modulated by BMI, suggesting a potential relationship between the rs7903146 variant and risk for obesity. Such factors may be related to the *TCF7L2* expression and the Wnt pathway regulation of adipose tissue (*Ross et al., 2000*; *Grant et al., 2006*; *Zhou et al., 2014*; *Cropano et al., 2017*; *Chen et al., 2018*).

The Wnt signaling pathways negatively regulate adipogenesis and play important metabolic and developmental roles in adipose tissue composition and functioning (*Chen & Wang, 2018*). Although the *TCF7L2* encodes the main effector involved in this signaling pathway, few studies investigate the association between the rs7903146 and risk for obesity (*Haupt et al., 2010*; *Al-Daghri et al., 2014*; *Locke et al., 2015*; *Abadi et al., 2017*; *Muller et al., 2019*). These studies reported an association between the rs7903146 C allele and the risk for obesity. In this sense, validation research for the association in other populations is compelling (*Grant, 2019*).

We hypothesize that the rs7903146 variant is associated not only with T2DM but also with obesity. We aimed to explore the relation between T2DM risk conferred by the rs7903146 SNP and BMI status, verifying differences among the rs7903146 genotypes on

PeerJ ___________________________________________________

BMI variation over a ten-year period. Furthermore, we performed interaction analyses of this genetic variant with BMI, age and gender.

## MATERIALS & METHODS

### Study cohort

We recruited elderly volunteers from a health survey called SABE (*Saúde, Bem Estar e Envelhecimento*—Health, Well-Being and Aging). SABE was carried out in São Paulo, Brazil, under the coordination of the Pan American Health Organization. The project was initially a multicenter health survey and well-being of older people in seven Caribbean and Latin American urban centers (Bridgetown, Barbados; Buenos Aires, Argentina; Havana, Cuba; Mexico City, Mexico; Montevideo, Uruguay; Santiago, Chile; and São Paulo, Brazil). The Brazilian center has since adopted a longitudinal approach with a new data collection every five years, under the coordination of the Public Health School at the University of São Paulo (*Lebrão & Laurenti, 2005*).

SABE was approved by the Ethics in Research Committee of the School of Public Health of the University of São Paulo and the Brazilian National Committee for Ethics in Research under protocol number 2015/12837/1.015.223. All participants signed an informed consent form under the Brazilian regulatory requirements of research with human subjects. A detailed description of the study population, including demographic characteristics, clinical and anthropometric data, medical history and socioeconomic background, has been previously published (*Lebrão & Laurenti, 2005*).

In addition, SABE was approved by the Institutional Review Board of the University of São Paulo School of Public Health (CAAE: 47683115.4.0000.5421, Review: 3.600.782). All genomic dataset subjects have agreed to participate in this study and signed the written informed consent form approved by CEP/CONEP (Brazilian local and national Ethical Committee Boards).

### Clinical and anthropometric characteristics

Data was collected by a specific standardized questionnaire (C10) proposed by the Pan American Health Organization (PAHO), which was translated and adapted for use in Brazil (*Naslavsky et al., 2017*). Trained interviewers administered the questionnaire in subjects' households. The T2DM was considered self-reported if the subject provided an affirmative answer to the question, "Have you ever been told by a doctor or other health professional that you have diabetes or high blood sugar levels?". Blood was withdrawn and submitted to biochemical and genomic analyses.

We assessed the following demographic and health variables: gender, age, fasting plasma glucose (mg/dL), glycated hemoglobin (%), total cholesterol (mg/dL), fasting triglyceride (mg / dL), LDL cholesterol (mg/dL), HDL cholesterol (mg/dL), systolic pressure (mmHg), diastolic pressure (mmHg), BMI ($kg/m^2$), waist circumference (cm), hip circumference (cm), and hip-waist ratio (cm/cm). All participants with T2DM and/or with blood glucose levels above 100 mg/dL were considered hyperglycemic. For anthropometric evaluation, the weight was measured using a portable scale (Seca, Germany), and the height was measured using an anthropometer (Harpenden, England).
Waist circumference was measured using an inelastic measurement tape placed on the midpoint between the lower margin of the last palpable riband and the top of the iliac crest. Hip circumference was measured around the widest portion of the buttocks.

The BMI was calculated from weight and height at baseline from the ratio of body weight (in kilograms) to height in square meters. We stratified individuals into three groups according to the World Health Organization (WHO) criteria for BMI classification: normal weight (18.5–24.9 kg/m$^2$), overweight (25.0–29.9 kg/m$^2$) and obesity (≥30.0 kg/m$^2$). Abdominal obesity was defined by the waist circumference >88 cm for women and >102 cm for men. Anthropometric data were analyzed twice over 10 years (2000 and 2010). The BMI variation (ΔBMI) of each elderly was obtained from the difference between the BMI measured in the collection years 2010 and 2000.

After excluding subjects with incomplete data, this study was carried out in a multiethnic population of 1,023 elderly individuals. The cohort included men and women whose anthropometric, biochemical, and genetic information were evaluated to verify association with T2DM, obesity, and BMI variation over ten years.

### Next-generation sequencing data

We selected the *TCF7L2* rs7903146 SNP because of its high predictive power (effect size × allele frequency) in the Latin American population. It is also localized in a gene related to various cellular processes involved in T2DM development (*Berumen et al., 2019*). We filtered the *TCF7L2* rs7903146 genotypes from the whole-genome sequencing dataset of SABE, the second phase of genomic analyses following the dataset deposited in ABraOM—*Arquivo Brasileiro Online de Mutações* (Online Archive of Brazilian Mutations, http://abraom.ib.usp.br). Quality control of genotypes and variants is described by *Naslavsky et al. (2017)* and by *Naslavsky et al. (2020)*.

### Statistical analysis

Data regarding continuous variables were expressed as percentages and the mean ± standard deviation (SD) and percentages for categorical variables. The one-sample Kolmogorov–Smirnov test was used to test the normality. Differences between groups for categorical data were tested by χ$^2$ analysis, while Independent Samples Mann–Whitney U test and the Kruskal–Wallis test were used for continuous data. Allele frequencies were determined by gene counting and Hardy–Weinberg equilibrium deviations were verified using a χ$^2$ test.

Allele and genotype distributions among groups were evaluated with χ$^2$ test or Fisher's exact test. The level of significance adopted was $P < 0.05$. Logistic regression models were developed after adjusting for age and gender and were performed to assess the independent role of the *TCF7L2* genotype. Interaction analysis was performed. The rs7903146 genotypic frequencies were compared among the ΔBMI tertiles. SPSS (version 25.0.0.0) software was used for general statistics.

A power analysis was performed using the software G*Power version 3.1.9.2 to verify the rs790314 association with T2DM and obesity. The sample size was 1,023, and to perform the power analysis were considered: a significance level of 0.05, the OR of 1.3,

**Table 1 Anthropometric and biochemical characteristics according to Body Mass Index status.**

| Variable | Unit | Normal-weight | Overweight | Obesity | P |
|---|---|---|---|---|---|
| Population size | N | 280 | 424 | 319 | – |
| Gender M/F | N/N | 115/165 | 183/241 | 67/252 | <0.0001 |
| Age | years old | 74.3 (65.8–82.6) | 71.6 (64.5–79.2) | 67.9 (64.0–75.2) | <0.0001 |
| BMI | kg/m$^2$ | 22.8 (21.0–24.0) | 27.3 (26.2–28.4) | 33.1 (31.2–35.9) | <0.0001 |
| Waist circumference | cm | 82.0 (77.8–87.0) | 94.0 (89.0–99.0) | 105.0 (100.0–110.0) | <0.0001 |
| Hip circumference | cm | 93.0 (90.0–96.0) | 101.0 (98.0–104.0) | 113.0 (108.0–120.0) | <0.0001 |
| Hip-waist ratio | cm/cm | 0.88 (0.83–0.93) | 0.93 (0.88–0.98) | 0.91 (0.87–0.97) | <0.0001 |
| Systolic pressure | mmHg | 134.3 (121.7–152.0) | 138.0 (127.7–153.0) | 138.0 (125.0–155.0) | 0.0664 |
| Diastolic pressure | mmHg | 76.2 (68.3–85.8) | 79.7 (72.0–86.3) | 81.0 (74.2–90.0) | <0.0001 |
| Plasma glucose | mg/dL | 85.0 (78.0–95.0) | 88.0 (81.0–102.3) | 93.0 (84.0–107.0) | <0.0001 |
| Glycated hemoglobin | % | 5.7 (5.5–6.0) | 5.8 (5.6–6.1) | 5.9 (5.6–6.3) | <0.0001 |
| Total cholesterol | mg/dL | 202.5 (177.0–234.5) | 200.0 (176.0–228.0) | 207.0 (180.0–230.5) | 0.3732 |
| Fasting triglyceride | mg/dL | 102.5 (75.0–137.3) | 116.5 (89.8–167.3) | 126.0 (94.0–168.5) | <0.0001 |
| LDL cholesterol | mg/dL | 126.0 (104.0–148.0) | 124.0 (104.8–149.0) | 130.0 (106.5–151.0) | 0.7908 |
| HDL cholesterol | mg/dL | 52.0 (42.8–62.0) | 45.0 (38.0–54.0) | 47.0 (41.0–56.0) | <0.0001 |
| T2DM | N (%) | 51 (18) | 117 (28) | 92 (29) | 0.0048 |
| Hypertensive | N (%) | 166 (59) | 279 (66) | 254 (80) | <0.0001 |

Notes:
Data are presented as median and range for the most variables; *P*-value with Kruskal–Wallis test for quantitative variables and Chi-square test for qualitative data.
BMI classification criteria: normal-weight (18.5–24.9 kg/m$^2$), overweight (25.0–29.9 kg/m$^2$), obesity (≥30.0 kg/m$^2$).
*P*, *P*-value; T2DM, type 2 diabetes mellitus; M/F, male/female; BMI, body mass index; LDL, low density lipoprotein; HDL, high density lipoprotein.

a statistical power of 90%, the expected squared coefficient of multiple correlations ($R^2$) of 0.25 (moderate association).

## RESULTS

The main clinical features of the 1,023 participants are depicted in Table 1. The median age of participants was 71.4 years old (59–99 years old), and 64.32% of participants were women. The three groups clustered by BMI status contain 280 subjects with normal weight, 424 with overweight and 319 with obesity (Table 1). These groups did not differ in systolic pressure, total cholesterol and LDL cholesterol. However, groups differed in gender ratio, age, waist circumference, hip circumference, hip-waist ratio, diastolic pressure, plasma glucose, glycated hemoglobin, fasting triglyceride, HDL cholesterol and number of subjects with T2DM and arterial hypertension (Table 1).

The genotypic distributions for the rs7903146 were in Hardy–Weinberg equilibrium in all groups (All *P* > 0.05). These genotypic and allelic distributions of participants with and without T2DM according to BMI status are shown in Table S1 (All *P* > 0.05). In addition, the analysis of ancestries frequency is demonstrated in Table S2. There was a significant difference in European contribution among the genotypes (*P* = 0.007) and between Non-T2DM and T2DM groups (*P* = 0.020).

The TT genotype was more frequent in the T2DM group (*P* = 0.0001). The *TCF7L2* rs7903146 T allele, in association with T2DM, was confirmed on the recessive genetic model (OR = 1.89; 95% CI [1.21–2.95]; *P* = 0.004), but no significant associations were

**Table 2 Association between rs7903146 TT genotype and risk for Type 2 diabetes adjusted for the confounding variables.**

| Possible confounding variable | *P*-value | Odds ratio | 95% Confidence interval |
|---|---|---|---|
| Age (years old) | 0.005 | 1.89 | [1.21–2.95] |
| Gender (*N*) | 0.005 | 1.90 | [1.22–2.97] |
| BMI (kg/m$^2$) | 0.003 | 1.95 | [1.25–3.06] |
| Waist circumference (cm) | 0.004 | 1.93 | [1.23–3.02] |
| Diastolic pressure (mmHg) | 0.006 | 1.88 | [1.20–2.93] |
| Glycated hemoglobina (%) | 0.004 | 2.34 | [1.31–4.18] |
| Fasting triglyceride (mg/dL) | 0.005 | 1.90 | [1.22–2.97] |
| HDL cholesterol (mg/dL) | 0.006 | 1.87 | [1.19–2.92] |
| European ancestrie (%) | 0.005 | 1.92 | [1.21–3.05] |
| All confounders toghether | 0.006 | 2.35 | [1.28–4.32] |

**Note:**
Regression logistic analysis was adopted.

detected with other phenotypes (Table S3). Significant association signals were detected between hyperglycemia and the T allele on the dominant genetic model (OR = 1.77; 95% CI [2.61–1.20]; *P* = 0.004) as well as on the log-additive model (OR = 1.56; 95% CI [2.09–1.17]; *P* = 0.002) (Table S4). Yet, a C allele protective effect was also observed against T2DM under dominant (OR = 0.51; 95% CI [0.32–0.80]; *P* = 0.003), additive (OR = 0.50; 95% CI [0.31–0.81]; *P* = 0.004) and allelic (OR = 0.79; 95% CI [0.64–0.98]; *P* = 0.031) genetic models (Table S5).

The regression analysis showed a risk for T2DM on TT carriers even after adjusting for all confounds depicted in Table 1 (Table 2). However, the interaction analysis demonstrated that BMI modifies the association between TT genotype and T2DM risk (OR = 1.02; 95% CI [1.01–1.04]; $P_{interaction}$ = 0.008). This result reinforces the logistic regression analysis stratified by BMI status, which showed that the risk for T2DM conferred by T allele is higher in the normal weight group, with the following odds ratios: (OR = 3.36; 95% CI [1.46–7.74]; *P* = 0.004) for the recessive model and (OR = 3.21; 95% CI [1.31–7.87]; *P* = 0.011) for the additive model (Table 3). In addition, the T allele and age interaction analysis demonstrated that the increased T2DM risk in TT carries is maintained, and the age affects this association (OR = 1.01; 95% CI [1.00–1.02]; $P_{interaction}$ = 0.005).

Association tests separated by gender showed a borderline association between the TT genotype and risk for T2DM in men (OR = 2.19; 95% CI [1.05–4.58]; *P* = 0.042). Tests also reported a trend for association in women (OR = 1.75; 95% CI [1.00–3.07]; *P* = 0.055). After grouping subjects from the normal weight and overweight groups and excluding the obese group, we observed association both in men (OR = 2.64; 95% CI [1.16–5.98]; *P* = 0.020) and women (OR = 2.14; 95% CI [1.08–4.21]; *P* = 0.028). However, in the normal weight group, we noticed a strong association in men (OR = 5.48; 95% CI [1.57–19.10]; *P* = 0.008) while there was no association in women. Furthermore, this result is also

**Table 3 Association of the rs7903146 T allele with type 2 diabetes mellitus according to Body Mass Index status.**

| BMI status | N | Dominant model (CC Vs CT + TT) | | Recessive model (CC + CT Vs TT) | | Additive model (CC Vs TT) | | Allelic model (C Vs T) | |
|---|---|---|---|---|---|---|---|---|---|
| | | OR (95% CI) | P | OR (95% CI) | P | OR (95% CI) | P | OR (95% CI) | P |
| Normal weight | 280 | 1.23 [0.66–2.27] | 0.512 | 3.36 [1.46–7.74] | 0.004 | 3.21 [1.31–7.87] | 0.011 | 1.52 [0.97–2.36] | 0.066 |
| Overweight | 424 | 1.20 [0.78–1.86] | 0.401 | 1.96 [0.99–3.87] | 0.054 | 1.98 [0.96–4.10] | 0.065 | 1.27 [0.92–1.74] | 0.141 |
| Obesity | 319 | 1.15 [0.71–1.88] | 0.565 | 1.23 [0.51–2.98] | 0.642 | 1.31 [0.53–3.28] | 0.560 | 1.13 [0.78–1.66] | 0.516 |
| Normal weight + overweight | 704 | 1.22 [0.86–1.73] | 0.276 | 2.31 [1.37–3.90] | 0.002 | 2.34 [1.34–4.08] | 0.003 | 1.34 [1.04–1.73] | 0.025 |
| Overweight + obesity | 743 | 1.16 [0.84–1.60] | 0.370 | 1.59 [0.93–2.72] | 0.089 | 1.65 [0.94–2.89] | 0.082 | 1.19 [0.94–1.52] | 0.155 |
| Total | 1023 | 1.16 [0.87–1.54] | 0.305 | 1.90 [1.22–2.97] | 0.005 | 1.94 [1.21–3.10] | 0.006 | 1.25 [1.01–1.54] | 0.042 |

Notes:
Volunteers without type 2 diabetes mellitus were considered as the control group.
BMI classification criteria: normal-weight (18.5–24.9 kg/m$^2$), overweight (25.0–29.9 kg/m$^2$), obesity ($\geq$ 30.0 kg/m$^2$).
P, P-value; 95% CI, 95% confidence interval; OR, odds ratio OR adjusted for sex and age.

**Table 4 Association of the rs7903146 C allele with the Body Mass Index Status.**

| Control group | Case group | Dominant model (TT Vs CC + CT) | | Recessive model (TT + CT Vs CC) | | Additive model (TT Vs CC) | | Allelic model (T Vs C) | |
|---|---|---|---|---|---|---|---|---|---|
| | | OR (95% CI) | P | OR (95% CI) | P | OR (95% CI) | P | OR (95% CI) | P |
| Normal-weight | Overweight | 1.24 [0.74–2.09] | 0.410 | 0.92 [0.68–1.26] | 0.611 | 1.15 [0.67–1.98] | 0.615 | 1.00 [0.79–1.26] | 0.984 |
| Normal-weight | Obesity | 1.60 [0.88–2.92] | 0.122 | 1.41 [1.00–1.98] | 0.052 | 1.73 [0.93–3.21] | 0.081 | 1.34 [1.03–1.75] | 0.029 |
| Normal-weight | Obesity + overweight | 1.32 [0.82–2.13] | 0.253 | 1.08 [0.81–1.43] | 0.604 | 1.31 [0.80–2.15] | 0.286 | 1.10 [0.89–1.36] | 0.373 |
| Normal-weight + overweight | Obesity | 1.29 [0.78–2.11] | 0.317 | 1.40 [1.06–1.84] | 0.017 | 1.48 [0.89–2.47] | 0.132 | 1.28 [1.03–1.58] | 0.024 |

Notes:
P, P-value; 95% CI, 95% confidence interval; OR, odds ratio OR adjusted for sex and age.
BMI classification criteria: normal-weight (18.5–24.9 kg/m$^2$), overweight (25.0–29.9 kg/m$^2$), obesity ($\geq$ 30.0 kg/m$^2$).

reinforced by the interaction analysis of the TT genotype and gender on the risk for T2DM (OR = 1.87; 95% CI [1.08–3.25]; $P_{interaction}$ = 0.026).

The association between the rs7903146 variant and obesity status was analyzed and the T allele conferred protection against obesity on the dominant model (OR = 0.71; 95% CI [0.54–0.94]; P = 0.016) (Table S6). This result leads us to believe that there is an association between the C allele and obesity. Our analysis revealed a CC genotype association with obesity risk on the recessive model (OR = 1.40; 95% CI [1.06–1.84]; P = 0.017) (Table 4). The regression analysis showed a higher risk for obesity on CC carriers even after adjustment for all the possible confounds (OR = 1.41; 95% CI [1.06–1.86]; P = 0.017) (Table 5). Additionally, we observed a borderline association with abdominal obesity in subjects with CC genotype (OR = 1.29; 95% CI [1.28–1.67]; P = 0.045).

Analysis of BMI variation over ten years of SABE study revealed a different distribution of rs7903146 genotypes among ΔBMI tertiles (Tables S7 and S8). We observed an increase in the TT genotype in the ΔBMI second tertile compared to the first tertile. The increased frequency in the TT genotype was detected on the recessive genetic model

**Table 5 Association between rs7903146 CC genotype and risk for obesity adjusted for the confounding variables.**

| Possible confounding variable | P-value | Odds Ratio | 95% Confidence Interval |
|---|---|---|---|
| Age (years old) | <0.001 | 0.96 | [0.95–0.98] |
| Gender (N) | <0.001 | 2.85 | [2.08–3.89] |
| Diastolic pressure (mmHg) | <0.001 | 1.02 | [1.01–1.03] |
| Glycated hemoglobina (%) | 0.032 | 1.13 | [1.01–1.27] |
| Fasting triglyceride* (mg/dL) | 0.034 | 1.00 | [1.00–1.00] |
| HDL cholesterol (mg/dL) | 0.443 | 1.00 | [0.99–1.00] |
| All confounders toghether | 0.017 | 1.41 | [1.06–1.86] |

Notes:
* Association between rs7903146 CC genotype and obesity adjusted by fasting triglyceride: $P = 0.034$; OR = 1.0016; 95% Confidence Interval = 1.0001–1.0031.
Regression logistic analysis was adopted.

(OR = 2.00; 95% CI [1.01–3.97]; $P = 0.044$) and in participants without T2DM, both on the additive (OR = 5.13; 95% CI [1.40–18.93]; $P = 0.009$) as on the recessive model (OR = 5.13; 95% CI [1.43–18.37]; $P = 0.010$) (Table S7). No significant values were found among individuals with T2DM (Table S8).

## DISCUSSION

We evaluated the *TCF7L2* rs7903146 association with T2DM and obesity. We explored whether the strength of association with T2DM depends on BMI status (normal-weight, overweight and obesity). The differences in BMI variation over ten years on C and T allele carriers were also investigated. We confirmed that the T allele risk confers risk for T2DM, and it is influenced by BMI status, age and gender. The TT genotype conferred a protective effect against obesity and the CC genotype was associated with risk for obesity. Moreover, the TT genotype was associated with a lower BMI variation over a 10-year period in our elderly cohort.

According to the Allele Frequency Aggregator (ALFA) project from the National Center for Biotechnology Information (NCBI) database, the worldwide frequency for the rs7903146 T allele is around 0.29 (*Phan et al., 2020*). In our population, T allele frequency varied from 0.27 to 0.33 among the categories, except for the normal weight diabetic elderly group. For this group, the T allele frequency was around 0.40 (Table S1). Thereafter, we performed regression analysis adjusted for gender and age, which confirmed the rs7903146 T allele risk for T2DM in the total population (Table 3).

Other Brazilian studies also reported the rs7903146 T allele association with risk for T2DM (*Barra et al., 2012*; *Assmann et al., 2017*). However, these studies did not investigate the influence of BMI on diabetes risk or addressed this issue considering an elderly population. Thus, we verified an increased risk for T2DM conferred by the rs7903146 T allele in lower BMI participants. This risk was higher in the normal weight group (OR = 3.36; 95% CI [1.46–7.74]; $P = 0.004$) (Table 3). Previous studies, with populations from other countries, also reported a higher risk for T2DM when the BMI is lower (*Cauchi et al., 2006*, *2008b*; *Bouhaha et al., 2010*; *Corella et al., 2016*). Cauchi et al. (2008) and

*Corella et al. (2016)* observed odds ratios of 1.89 (95% CI [1.67–2.14]) and 2.32 (95% CI [1.90–2.85]) in individuals without obesity, respectively. Meanwhile, *Bouhaha et al. (2010)* reported an odds ratio (OR = 3.24; 95% CI [1.10–9.53]) similar to our results. *Perry et al. (2012)*, when analyzing 36 diabetes loci, verified, in 29 of them, a higher risk for T2DM in normal weight individuals compared to obese individuals.

The rs7903146 T allele may have a more significant impact on individuals without obesity. This impact is due not to the obesity-induced insulin resistance but due to pancreatic dysfunction, indicating that β-cell impairment predicts a future T2DM in subjects with lower BMI (*Cauchi et al., 2008b*; *Bouhaha et al., 2010*). Among leaner subjects, the β-cell compensation is lower, while among people with obesity, the compensation is higher (*Watanabe et al., 2007*). Plasmids carrying the T allele showed more robust transcriptional activity when compared to those with the C allele. In addition, pancreatic cells with T allele carriers showed impaired proinsulin processing, resulting in a high level of proinsulin in the plasma and an increased proinsulin/insulin ratio (*Stolerman et al., 2009*). Human islets have a higher degree of open chromatin, corroborating that the T allele leads to an increased expression of *TCF7L2* and decreased insulin content and secretion (*Zhou et al., 2014*). Additionally, *Zhou et al. (2014)* demonstrated that in islets from CC genotype carriers, *TCF7L2* mRNA expression was negatively associated with the genes *ISL1*, *MAFA* and *NKX6.1* but not with *MAFA* and *NKX6.1* in CT/TT genotype carriers. This finding reinforces the β-cell impairment in T allele risk carriers.

The difference in the association of genes between men and women has been previously explored. Interestingly, there is evidence that the *TCF7L2* gene behaves differently in women and men. Moreover, *He, Zhong & Cui (2014)* conducted an integrated approach and observed gender differences in association signals at the gene-level and pathway-level. In this study, the *TCF7L2* association was found only in male subjects, and all SNPs in this gene, considering the female population, did not show significance. Since *TCF7L2* belongs to several enriched pathways and is widely recognized as a gene conferring risk of T2DM, the authors performed the same analysis but deleting this gene in all pathways. They observed no significant change between pathway signals with and without the *TCF7L2* gene in the female group, while the strong signals in the male group were almost nonexistent after deleting the gene. This evidence suggests a potential difference in T2DM etiology in the pathway-level for each gender group. The authors verified that the significance of the pathways in the male group is primarily dominated by the *TCF7L2* gene (*He, Zhong & Cui, 2014*).

*Berumen et al. (2019)* investigated the influence of several factors on T2DM variability in Mexico. The authors observed that the factors contributed more in men (33.2%) than in women (25%). In addition, genes played a substantially more important role in men than in women (14.9% vs. 5.5%), while obesity and parental history played a similar role in both genders. Genes and parental history appeared to play a more significant role than obesity in T2DM. According to Berumen et al., the effect of *TCF7L2* on men is more significant when the disease is diagnosed at ≤45 years of age than when it is diagnosed at an older age, especially for the homozygous risk allele (OR = 4.62 and 2.59, respectively).

Thus, the previously mentioned studies could explain the interaction between the T allele and gender on diabetes risk, which was observed in our study.

Our selected gene variant represents only a fraction of the studied gene's potential variation and the mechanisms involving *TCF7L2*, T2DM and obesity. Additional genetic studies with other variants are needed better to understand the *TCF7L2* role in these complex diseases. The rs12255372 variant in intron 4 of the *TCF7L2* gene showed strong linkage disequilibrium (LD) with rs7903146 (*Pang, Smith & Humphries, 2013*). Moreover, subjects homozygous for the risk-associated allele showed higher gene expression in pancreatic islets and were more than twice as likely to develop T2DM as non-carriers (*Lyssenko et al., 2007*; *Pang, Smith & Humphries, 2013*).

Prior association studies reported a lack of association between the rs7903146 T allele and obesity status (*Cauchi et al., 2008b*; *Stolerman et al., 2009*; *Bouhaha et al., 2010*; *Al-Safar et al., 2015*). However, we verified a T allele protective effect against obesity (Table S6), corroborating results reported in more recent studies (*Noordam et al., 2017*; *Fernández-Rhodes et al., 2018*). A cross-sectional analysis conducted in middle-aged participants (mean age of 55.9 ± 6.0 years) reported a T allele association with lower BMI and mean total body fat (*Noordam et al., 2017*). Furthermore, *Fernández-Rhodes et al. (2018)* showed an association between TT genotype, decreased waist circumference and lower mean BMI at multiple time points in the life course. This protection against obesity might be due to reduced insulin production and secretion related to the rs7903146 T allele once insulin stimulates the increased glucose uptake in adipocytes. Insulin plays, therefore, a pro-obesogenic role both from its anabolic effect on lipid accumulation and due to compensatory eating to prevent episodes of hypoglycemia (*Zhou et al., 2016*).

Multiple factors are related to the changes in body composition with aging. From the fourth decade onwards, the muscle mass declines and accounts for reduced resting metabolic rates, which contribute to the gradual increase in body fat in elderly subjects (*Gallagher et al., 1998*; *Sayer et al., 2008*). Around 75 years of age, the BMI appears to be stable, yet it is overestimated due to an increase in fat mass and a decrease in lean mass and bone density (*Ponti et al., 2020*). In this sense, due to sarcopenic obesity, it is not easy to differentiate lean and obese elderlies. BMI cutoff points are still controversial for this range of age and, therefore, BMI classification is a limiting factor for our cohort. However, the BMI variation is a substantial risk predictor for elderlies and the rs7903146 T allele protective effect against obesity deserves attention—especially since thinness is a significant risk factor in old age, and weight loss is closely related to the frailty syndrome and other health complications (*Aune et al., 2016*; *Di Angelantonio et al., 2016*; *Ponti et al., 2020*).

Studies have reported clinical implications of sarcopenic obesity in subjects with T2DM (*Khadra et al., 2019*; *El Ghoch, Calugi & Grave, 2018*; *Kim & Park, 2018*). A recent meta-analysis observed that the presence of sarcopenic obesity increases the T2DM risk by 38% concerning those without sarcopenic obesity (OR = 1.38, 95% CI [1.27–1.50]) (*Khadra et al., 2019*). The commonly accepted mechanism interconnecting T2DM and sarcopenic obesity involves an increase in fat mass, decrease in lean mass, chronic inflammation and insulin resistance; however, the mechanism itself is still unclear (*Ponti*

et al., 2020). It can thus be said that the interaction between the T allele and age on diabetes risk observed in our elderly cohort could be related to the sarcopenic obesity in older adults and the age-related decline in resting metabolic rates.

Our data further suggest a differential effect of rs7903146 genotypes in BMI variation only in elderly subjects without T2DM (Table S7). Minor variation in BMI was observed among the TT genotype during the SABE 10-year period. This conclusion is drawn from the increased number of TT subjects without T2DM on the second tertile of ΔBMI values (Table S7). The same result is found in interventional studies that verified lower BMI variation in rs7903146 T allele carriers (Haupt et al., 2010; Kaminska et al., 2012; Roswall et al., 2014). Mattei et al. (2012) observed a greater loss of lean mass for CC subjects on a low-fat diet compared to TT (Mattei et al., 2012). Similarly, less weight gain per year was observed in patients with the T allele compared to the C allele after adopting a Mediterranean diet (Roswall et al., 2014). According to Fisher et al. (2012), the rs7903146 C allele arose during the transition from hunter-gatherer to agricultural practices, with a consequent reduction of protein sources. Carriers of the rs7906146 T allele were then selectively adapted to maintain weight stability under low-protein conditions (Fisher et al., 2012).

Helgason et al. (2007) reported that the rs7903146 T allele was the probable ancestral allele, serving for a better subjacent mutation. In addition, the authors identified a haplotype with the C allele (HapA). This haplotype indicates positive selection, besides the association with BMI and altered concentrations of ghrelin and leptin. It further indicates that the selective advantage of HapA may have been mediated through effects on energy metabolism (Helgason et al., 2007). Corroborating with these findings, the most significant GWAS meta-analysis for BMI so far (~300,000 subjects) reported a C allele association with BMI (Locke et al., 2015). Although the extent of clinical variability associated with the C allele is not fully known, significant associations between the rs7903146 C allele and BMI and/or waist circumference were observed in a Saudi population (Al-Daghri et al., 2014), in European adults (Abadi et al., 2017) and American Indians (Muller et al., 2019).

CC genotype subjects expressed more transcripts containing the alternative spliced exons (13 and 13a) in their adipose tissue associated with BMI and body fat percentage than the T allele carriers (Kaminska et al., 2012). Furthermore, in addition to nine diabetes-associated genes, five in seven TCF7L2 splice forms were differentially expressed by comparing leukocyte cells of carriers of the CC and CT/TT genotypes. This ratio might reflect a significant change in gene interactions and responsible networks as glucose homeostasis, adipogenesis and others (Vaquero et al., 2012). In this sense, the TCF7L2 alternative splicing in the adipose tissue could be regulated by health, disease, weight loss and insulin resistance (Mondal et al., 2010; Kaminska et al., 2012; Vaquero et al., 2012; Zhou et al., 2014; Chen et al., 2018).

The TCF7L2 gene plays important metabolic and developmental roles in adipose tissue composition and functioning. Therefore, it is largely hypothesized that the Wnt signaling is critical for obesity development (Chen & Wang, 2018; Chen et al., 2018). This gene is differentially methylated in adipose tissue, exhibiting relevant epigenetic changes to the

development of both diabetes and obesity (*Nilsson et al., 2014*). The *TCF7L2* protein inactivation is associated with increased subcutaneous adipose tissue mass, adipocyte hypertrophy and inflammation (*Chen et al., 2018*). Furthermore, besides alternative splicing, other regulatory changes seem to be genotype-specific and further affect the *TCF7L2* role in adipose tissue. Several protein factors, including GATA3, a transcription factor that controls the preadipocyte-to-adipocyte transition, bind only to the rs7903146 C allele but not to the T allele under caloric restriction (*Cauchi et al., 2008a*).

This evidence supports the CC genotype association with the risk of obesity and abdominal obesity in our population. Thus, we speculate that the inverse effects observed in our study of the rs7903146 T and C alleles on risks for diabetes and obesity. Such effects might be related to the *TCF7L2* expression and its genotype-specific effects on the WNT signaling pathway in adipose tissue and others. We recognize the advances in knowledge regarding the production, processing, trafficking and secretion of insulin. However, more studies are required to understand further the mechanisms interconnecting the *TCF7L2* rs7903146 variant, T2DM and obesity.

The main strength of this study is the median age of our population, which exceeds the age of onset of diabetes and obesity, thus minimizing a typical bias in the selection of the control group. As far as we know, this is one of the few association studies that reported an association of the rs7903146 variant with BMI variation during a 10-year-interval assessment. It is also unique to investigate obesity status in an exclusively elderly cohort. Dietary factors play an essential role in T2DM etiology and the gene-diet interaction could influence T2DM pathogenesis (*Ouhaibi-Djellouli et al., 2014*; *Hindy et al., 2016*). Therefore, the lack of assessment regarding dietary aspects as well as physical activity levels could be a limitation of our study. However, we could detect and confirm the association between T2DM and the rs7903146 T allele in our population, worldwide recognized as the strongest GWAS signal for diabetes risk (*Grant, 2019*). Despite our population size, we were able to reproduce significant results following more recent studies performed on larger populations (*Locke et al., 2015*; *Abadi et al., 2017*; *Fernández-Rhodes et al., 2018*).

## CONCLUSIONS

We confirmed that the rs7903146 variant is associated with both T2DM and obesity. This result is supported by evolutive aspects and functional studies concerning the T and C alleles, contributing to knowledge expansion regarding this barely explored association. In addition, we found a TT association with a lower BMI variation in elderly subjects over the ten years of the SABE study. These findings provide a unique contribution to the field of association studies about this polymorphism. Nevertheless, additional studies are needed to understand the *TCF7L2* rs7903146 association with obesity and with BMI variation in different age groups of populations across the world.

## ACKNOWLEDGEMENTS

The authors acknowledge all volunteers and professionals who participated in the SABE survey. They would also like to thank the Academic Publishing Advisory Center (*Centro de*

*Assessoria de Publicação Acadêmica*, CAPA) of the Federal University of Paraná (UFPR) for assistance with English language editing.

### Funding

This work was supported by INCT/FAPESP via Research, Innovation and Dissemination Centers (2014/50931-3), FAPESP/CEPID (2013/08028-1), National Council for the Development of Science and Technology (CNPq-Casadinho-Procad/Edital 06, Ref. 552672/ 2011-4/MCTI/MEC/CAPES), and CAPES for Lais Bride's scholarship.
The funders had no role in study design, data collection and analysis, decision to publish, or preparation of the manuscript.

### Grant Disclosures

The following grant information was disclosed by the authors:
INCT/FAPESP via Research, Innovation and Dissemination Centers: 2014/50931-3.
FAPESP/CEPID: 2013/08028-1.
National Council for the Development of Science and Technology: 552672/ 2011-4/MCTI/ MEC/CAPES.
CAPES, FAPES and Federal University of Espírito Santo (Fundo de amparo à pesquisa linha IV – UFES).

### Competing Interests

The authors declare that they have no competing interests.

### Author Contributions

- Lais Bride conceived and designed the experiments, performed the experiments, analyzed the data, prepared figures and/or tables, authored or reviewed drafts of the paper, and approved the final draft.
- Michel Naslavsky conceived and designed the experiments, performed the experiments, analyzed the data, authored or reviewed drafts of the paper, and approved the final draft.
- Guilherme Lopes Yamamoto performed the experiments, analyzed the data, authored or reviewed drafts of the paper, and approved the final draft.
- Marilia Scliar performed the experiments, analyzed the data, authored or reviewed drafts of the paper, and approved the final draft.
- Lucia HS Pimassoni conceived and designed the experiments, performed the experiments, analyzed the data, prepared figures and/or tables, authored or reviewed drafts of the paper, and approved the final draft.
- Paola Sossai Aguiar conceived and designed the experiments, performed the experiments, analyzed the data, authored or reviewed drafts of the paper, and approved the final draft.
- Flavia de Paula performed the experiments, analyzed the data, authored or reviewed drafts of the paper, and approved the final draft.

- Jaqueline Wang performed the experiments, analyzed the data, authored or reviewed drafts of the paper, and approved the final draft.
- Yeda Duarte conceived and designed the experiments, performed the experiments, authored or reviewed drafts of the paper, and approved the final draft.
- Maria Rita Passos-Bueno conceived and designed the experiments, authored or reviewed drafts of the paper, and approved the final draft.
- Mayana Zatz conceived and designed the experiments, authored or reviewed drafts of the paper, and approved the final draft.
- Flávia Imbroisi Valle Errera conceived and designed the experiments, analyzed the data, prepared figures and/or tables, authored or reviewed drafts of the paper, and approved the final draft.

### Human Ethics

The following information was supplied relating to ethical approvals (i.e., approving body and any reference numbers):

The SABE Study was approved by the Institutional Review Board of the University of São Paulo School of Public Health (CAAE: 47683115.4.0000.5421, Review: 3.600.782). All participants signed the free and informed consent form.

### Data Availability

Individual-level raw data cannot be publicly shared due to IRB restrictions. Aggregate data is available at http://abraom.ib.usp.br/ and individual-level genotype and phenotypic data can be shared upon reasonable request and approval of the research collaboration agreement. To request these data, please contact the database administrator at abraom@ib.usp.br.

Sequencing data:

The WGS data is available in Naslavsky et al.'s supplemental file at the link: DOI 10.1101/2020.09.15.298026.

### Supplemental Information

Supplemental information for this article can be found online at http://dx.doi.org/10.7717/peerj.11349#supplemental-information.

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
