# Peer review of "TCF7L2 rs7903146 polymorphism association with diabetes and obesity in an elderly cohort from Brazil"

_PeerJ, doi:10.7717/peerj.11349_

## Round 0.1 · original submission · Major Revisions

The reviewers have found scientific merit in your work, but there are some issues which you should address in a revised version of the text. Please, see their reports below this letter.

Reviewer 1 ·

Basic reporting

Bride et al examined the TCF7L2 rs7903146 (C>T) association with T2D and obesity in a ten year cohort of 1,023 elderly Brazilians. They found the T allele increased T2D risk in the normal-weight group. The C allele increased the obesity risk, while the T allele showed protective effect against obesity.
The manuscript is clear and well-written, and the statistical analyses were performed appropriately. Appropriate references are cited.

Experimental design

Research questions are well-defined, methods and experimental approach are clear with sufficient detail and information to replicate.

Validity of the findings

Conclusions are based on the findings.

Additional comments

1. The T2D group was diagnosed based on respons to questions “has a doctor or nurse ever told you that you have diabetes or high146 blood sugar levels?” Have the authors confirmed the T2D grouping based on the result of fasting plasma glucose and glycated hemoglobin measurement?
2. The authors found that based on logistic regression analysis, the risk for T2D varied according to BMI status (Results line 217). The pathophysiology of obesity is different in male and female. Did the authors observed any difference in T2D risk in male vs. female? Could the authors please do separate analysis for male and female, and compare the results?
3. The median age is significantly different among the 3 groups (normal-weight vs. pre-obesity vs. obesity). Could the authors kindly elaborate whether their finding that the T allele increased T2D risk in the normal-weight group was not due to the older age of this group? Has the author performed interaction analysis of the T allele and age in increasing the T2D risk?
4. The authors described an increase in the TT genotype in the ΔBMI second tertile when compared to the first tertile in Supplemental Table 3 “Genotypic distributions by tertile intervals from ∆BMI values of volunteers without type 2 diabetes mellitus”. Does this increment also observed among subjects with T2D?
5. Dietary factors such as fiber (Hindy et al. Genes & Nutrition. 2016. 11:6 doi: 10.1186/s12263-016-0524-4) and dessert (Ouhaibi-Djellouli et al. BMC Genet. 2014. 15:134. doi: 10.1186/s12863-014-0134-3) intakes could also play an important role in T2DM etiology, and gene-diet interaction could be present in T2DM pathogenesis. Therefore, it would be appropriate if the authors could include the lack of assessment of dietary intakes as one of the limitation of their study.

Minor comments:

1. The sentence in Discussion line 242-244 is not clear, could the authors please rephrase their sentence: “We confirmed that the genetic susceptibility conferred by rs7903146 to T2DM is influenced by BMI status and the TT genotype, at risk for T2DM, is associated with a protective effect against obesity.”
2. The sentence in Discussion line 295 is not clear, could the authors please clarify what they mean: “...and it difficult the recognition of eutrophic versus obesity status.”
3. Sentence structure in Discussion line 299-300 need to be revised: “In elderlies without T2DM, was observed an increased number of TT genotype carriers...”

Annotated reviews are not available for download in order to protect the identity of reviewers who chose to remain anonymous.

Reviewer 2 ·

Basic reporting

According to the WHO definitions, use "overweight" instead of "pre-obesity" to define the group with BMI of 25.0–29.9 kg/m2.

Experimental design

Sample size should be added (based on the expected prevalence of this SNP in Brazilians/closest population).

Validity of the findings

No Comment.

Additional comments

Thank you for submitting your interesting study. Please comment on the other TCL7 gene variants that may be in linkage disequilibrium with the one in your study, or maybe significant in obesity, metabolic syndrome, and/or Diabetes mellitus. Only a few editing points and language final check are required.

Reviewer 3 ·

Basic reporting

This manuscript describes the association with genetic variation in the TCF7L2 gene and diabetes and obesity in a Brazilian population. This is a relevant study and overall clearly written. However, before this manuscript can be accepted, there are still some concerns to be resolved regarding the rationale of this study as well as the description and interpretation of the results.

• My biggest concern is that it is not clear what exactly the hypothesis is that is being tested here. Is the question whether the association with TCF7L2 varies per BMI group, or is the question how strong the association is between TCF7L2 and BMI? This needs to be clarified and the manuscript needs to be adjusted accordingly.

• Table 2 provides an overview of the population characteristics per genotype, but an odds ratio is discussed in the text. I actually expected a table here that would show an effect estimate for the association with T2DM for both genotypes. Perhaps it could be considered to add such a table. This is because the current table feels more like a second Table 1 (this can be transferred to the supplement), instead of the table with research findings. In addition, it is remarkable that only two genotype groups instead of all groups are listed here.

• In Table 3 the description of the different models is different than in Table 2. For example CC + CT is now called the Dominant model. This is confusing. It would be better to be more consistent throughout the paper.

• Why is the association with the CC gene and obesity discussed? What is the hypothesis here?

• It is not entirely clear why the distribution of genotypes in the BMI variation groups is looked at instead of the strength of the association with T2DM. What is the hypothesis here? What is the rationality to look at BMI in subgroups based on BMI? This is not clearly described.

• Consider rewriting the sentence 242 as this is unclear.

Experimental design

• From line 188 it is discussed how the groups are formed based on the genotype. It is not entirely clear why these groups were chosen. In addition, the sample of this study is quite small and I would like to see how many individuals there are per group. Why was a continuous model not chosen?

• The three groups formed here differ substantially from each other in risk factors associated with obesity. This is plausible since the groups are based on this. However, this is not sufficiently highlighted in the section describing Table 1. This should be added.

• From the results of Table 3 it can be deduced that there is an association between rs7903146 and T2DM in the normal weight group, but not in the overweight groups. This is difficult to understand and looking at the confidence intervals, the groups could be very small. Please add the number of people per group as well as the number of cases and controls to this table.

• As far as I understand correctly, there is no adjustment for possible confounders. I understand that these are genetic associations, but it is still common to adjust for age and sex (and principal components of the study). Especially since there are differences in these variables between the groups and the strong correlation with the outcome. My advice would be to add a model that is adjusted for age and sex, and a model that is adjusted for all the confounders mentioned in Table 1. This gives a better picture of the association and how it is driven.

• It is a consideration to add a formal interaction with obesity. I would like to know what the p for interaction for BMI is in the association of rs7903146 and T2DM. Moreover, this will give a greater power than the stratification that is currently used.

Validity of the findings

• In the discussion it is discussed that an association with T2DM is only found in the group with lower BMI and possible biological mechanisms are discussed. However, the reason for the absence of an association in subjects with a high BMI is not sufficiently discussed. Could it be that BMI is such a strong factor that the genetic aspect is overruled here and BMI is therefore a stronger risk factor for T2DM than genetic variation in the TCF7L2 gene?

• The discussion speaks of minimizing selection bias because an elderly population has been chosen. However, the normal-weight and obesity group do differ based on age and it could be that there is survival bias in these analyses. Please elaborate in more detail on this and possibly give a solution for this in the analyses (adjust for age?).

Additional comments

• This is a multiethnic cohort. Please provide numbers of the different ancestries included in this cohort.

• There are several minor grammatical errors in the text. When referring to a model it is better to use 'for' instead of 'on'. See for example rule 219 'on the recessive model' could be better written as 'for the recessive model'. Moreover, 'the' is frequently used, although this is not always necessary. See for instance 220 where 'the obesity' can be better written as 'obesity'. These minor grammatical errors are more common in the paper, please check. In the discussion there are more grammatical errors, please reread and rewrite the manuscript where appropriate.

---

## Round 0.2 · Minor Revisions

Still pending a minor change to be assessed in a new revised version of the text.

Reviewer 1 ·

Basic reporting

The revised manuscript is much clearer.

Experimental design

Methods and experimental design are clear and sufficient.

Validity of the findings

Conclusions were drawn based on the findings.

Additional comments

All of this reviewer’s concerns have been addressed appropriately by the authors.

Reviewer 2 ·

Basic reporting

Further language improvement/ editing is required.

Experimental design

The author did not calculate the sample size. Software is available (e.g. g power). They need, at least, to explain why they included 1,023 patients in their study.

Validity of the findings

OK

Additional comments

None

---

## Round 0.3 · accepted · Accept

All the reviewers' concerns have been correctly addressed.

Reviewer 2 ·

Basic reporting

The submitted manuscript describes the association of TCF7L2 rs7903146 polymorphism with diabetes and obesity in an elderly cohort from Brazil.

Experimental design

Previous reviewers' comments were addressed, including the calculation of the sample size.

Validity of the findings

Previous reviewers' comments were addressed

Additional comments

Previous reviewers' comments were addressed. No further comments to be added from my side.

---

## Author Rebuttal · Round 0.3

Biotechnology Graduate Program
Federal University of Espírito Santo
Avenida Marechal Campos 1468 Maruipe
Vitória ES 29075-910
Tel: +55 27 3335 7447
http://www.biotecnologia.ufes.br/

March 25th, 2021

To Editor

Antonio Palazón-Bru

Peer J

Dear Editor,

We are sending the new version of the manuscript (Article ID 53366) including, as requested, the sample power calculation (line 203 to 207). We also included a new paragraph between lines 857 to 880 in discussion session, and checked all tables. The English language correction was performed (certificate attached) and the following title was suggested *"TCF7L2* rs7903146 polymorphism association with diabetes and obesity in an elderly cohort from Brazil".

I would like to thank you for the opportunity to improve the manuscript.

Yours sincerely,

Dr Flavia Errera
Corresponding Author
Associate Professor of Department of Biological Sciences
Tel: +5527999195218

E-mail: flavia.valle@ufes.br; flavia.errera@gmail.com

Universidade Federal do Espírito Santo –  Departamento de Ciências Biológicas

Av. Fernando Ferrari 514, Goiabeiras, sala 105. Vitória/ES, Brasil. CEP 29075-910

Este documento foi assinado digitalmente por FLAVIA IMBROISI VALLE
Para verificar o original visite: https://api.lepisma.ufes.br/arquivos-assinados/164500?tipoArquivo=O

[Figure]

UNIVERSIDADE FEDERAL DO ESPÍRITO SANTO

**PROTOCOLO DE ASSINATURA**

[Figure]

O documento acima foi assinado digitalmente com senha eletrônica através do Protocolo Web, conforme Portaria UFES nº 1.269 de 30/08/2018, por
FLAVIA IMBROISI VALLE - SIAPE 2305782
Departamento de Ciências Biológicas - DCB/CCHN
Em 25/03/2021 às 18:09

Para verificar as assinaturas e visualizar o documento original acesse o link:
https://api.lepisma.ufes.br/arquivos-assinados/164500?tipoArquivo=O

Este documento foi assinado digitalmente por FLAVIA IMBROISI VALLE
Para verificar o original visite: https://api.lepisma.ufes.br/arquivos-assinados/164500?tipoArquivo=O